# Tail Prediction for Heterogeneous Data Center Clusters

**Sharaf Malebary** [1,*] **, Sami Alesawi** [2] **and Hao Che** [3]

1   Department of Information Technology, Faculty of Computing and Information Technology, King Abdulaziz University, Rabigh 21911, Saudi Arabia
2   Department of Computer Science, Faculty of Computing and Information Technology, King Abdulaziz University, Rabigh 21911, Saudi Arabia
3   Department of Computer Science and Engineering, University of Texas at Arlington, Arlington, TX 76010, USA
*   Correspondence: smalebary@kau.edu.sa

**Abstract:** Service providers need to meet their service level objectives (SLOs) to ensure better client experiences. Predicting tail sojourn times of applications is an essential step to combat long tail latency. Therefore, as an attempt to further unravel the power of our prediction model, new study scenarios for heterogeneous environments will be introduced in this research by using either of two methods: white- or black-box solutions. This research presents several techniques for modeling clusters of inhomogeneous nodes. Those techniques are recognized as heterogeneous fork-join queuing networks (HFJQNs). Moreover, included in the research is a nested-event-based simulation model, borrowing help from multi-core technologies. This model adopts the multiprocessing technique to take part in its design to enable different architectural designs for all computing nodes. This novel implementation of the simulation model is believed to be the next logical step for research studies targeting heterogeneous clusters in addition to the several provided scenarios. Experimental results confirm that even with the existence of such heterogeneous conditions, the tail latency can be predicted at high-load regions with an approximated relative error of less than 15%.

**Keywords:** tail prediction; heterogeneous fork-join queuing networks; inhomogeneous nodes





## 1. Introduction

The unacceptable performance of many *user-facing* and *online data-intensive (OLDI)* services, at commercial data centers *(DCs)*, revealed an emerging desire to develop smart, lightweight, and fast scheduling algorithms. This unveiled interest dictates a requirement to have an additional module devoted to predicting request tail latency, where (1) applications tail SLOs translated into budgets, and (2) are used to compare and secure proper resources ahead of time. Principally, this dire need to predict long tails out of distributions of latency continues due to the sheer amount, and dynamic workload demands result in high variability within task *response times (RTs)*. Nowadays, many application requests are poured into a common heterogeneous infrastructure shared among applications. Therefore, heterogeneity in resources is also known to be one of the leading causes of the high variability among task response times. To this point, as a reason to mitigate the high latency and to improve the decision-making for scheduler(s), many earlier scheduling algorithms and an abundance of engineered techniques have been introduced [1–3]. Unfortunately, most of these scheduling designs, if not all, were based on a homogeneous assumption which renders their inability for adoption as practical solutions for heterogeneous environments. As a result, it becomes one of the principal causes of the expected poor user experience for all proposed solutions.

Consolidating cloud ecosystems helps reduce costs and improve the return on investment by sharing cluster resources at data centers. Enforcing such a practice would increase resource utilization, but unfortunately, it comes with its pitfalls. On one hand, it introduces a significant departure from using organization-specific clusters. For example, many

machines with various generational specifications and computing capacities are naturally used to satisfy the diversity in application demands. On the other hand, consolidated applications typically need to run their daemons or other software components in the backgrounds of computing units. Sometimes, executing these software components lead to an increased amount of different process hiccups at DCs. Hiccups differ concerning their duration or the additional hardware characteristics that might be required by daemons to operate. Further, contention over resources, either between applications or among the same application requests, maintenance activities, hot spots, energy management, and thermal and power effects are all pitfalls that contribute to creating various inhomogeneous conditions [4–6]. Therefore, when non-homogeneous situations exist, providing low latency services at scale is considered most important to many service providers.

The emerging dilemma has challenged lots of researchers to offer better ways to keep the tail of latency distribution as short as possible for latency-sensitive services [7–11]. So far, none of the presented works were thoughtfully designed to take into account users' wishes to maintain the promised guaranteed services. The leading cause is the missing link between the system-level requirements and the subsystem-level performance. Fortunately, we presented in previous publications a prediction model, which serves to compensate for the missing baseline connection. The model has the potential to work as a universal answer for all types of DC infrastructures, either shared or not [12].

The design space for creating the before-mentioned sophisticated and diverse DC climates, at first glimpse, seems endless and very challenging. Still, it is possible to establish simplified models for the study of this research. For example, fork-join structures are used generally to express the basic functional system at DCs, as they have become the core foundation for such environments. Fork-join structures are traditionally modeled by a class of queuing network models known as *fork-join queuing networks (FJQNs)*. The focus, mainly, will be on enabling computing units independently to demonstrate distinct responses when servicing a received group of tasks. Hence, by creating such heterogeneous FJQNs behaviors, it is possible to mimic what happens in production environments. This can be easily done by providing several case scenarios of inhomogeneous situations linked to many examples from reality.

However, the lack of establishing inhomogeneous conditions in all previously researched models, particularly regarding the prediction of tail latency, has become the downside that is reflected. Therefore, it is believed that the outcome of this work might pioneer future studies of tail prediction concerning inhomogeneous conditions. Thus, the prime goal is to demonstrate those complex HFJQNs using model-driven simulations Figure 1. In addition, a mathematical explanation emphasizes the white-box approach, which is a closed-form solution for any model whose analytical expressions for the means and variances of task response times are available, and comparisons between the two methods of prediction—white- and black-box—will be drawn. Finally, as a contributing part of this research, a mathematical analysis of the function of the budget translating unit will be presented. This analysis will describe how to translate requested SLO requirements into tasks' budgets to reserve proper resources ahead of time.

As explained, the outcomes of this research might become a real treasure for researchers interested in conducting further investigations utilizing the fork-join queuing models related to subjects of parallel processing. The lack of enough studies for tail prediction in inhomogeneous environments is the main reason for our work. There are three new novel ideas that contributed to this research in the late sections: (1) covering more advanced case studies of heterogeneous clusters, (2) convergence point for reliable metrics, and (3) budget translation, which renders the uniqueness of this work, as it takes further steps toward providing a practical, reliable, and budget guaranteed scheduling mechanisms in the foreseeable future. Therefore, our research work is formulated as the following: The related work is given in Section 2. In Section 3, a general overview of the derived prediction model and a closed-form solution for any inhomogeneous white-box models, in general, is given. In Section 4, we present abstract definitions of how to model

inhomogeneous conditions in fork-join structures and list all possible case scenarios of the defined in-homogeneous types, and it is followed by a discussion about the results of the experiments in Section 5. The convergence point for reliable metrics is described in Section 6. In Section 7, we give our explained method for budget translation. Finally, the future work and conclusion are given in Section 8.

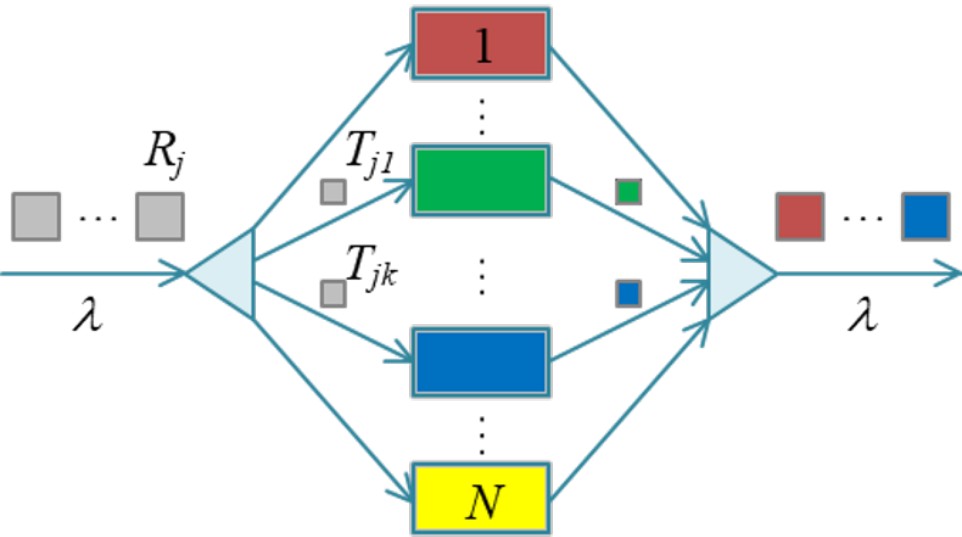

**Figure 1.** Simulation model for inhomogeneous scenarios, where incoming requests spawned to an equal number of fork nodes at a cluster.

## 2. Related Work

Fork-join structures underlay many big data applications, whose execution usually is handled by a large number of nodes, and the partial results from those nodes are then merged. Fork-join structures are traditionally modeled by fork-join queuing networks (FJQNs) [13]. FJQN models have been studied extensively in the literature. To date, the exact solution exists for a two-node network only [14,15]. Most of the previous research efforts mainly focus on approximating the mean response time for the FJQN model [15–17] and its bounds [18–20]. Some works attempt to find the approximation of response time distribution for the networks applying simple queuing models for each node, e.g., M/M/1 [21] or M/M/k [22], i.e., Poisson arrival process and exponential service times with one or more servers. For general service time distribution, the combination of analysis and simulation are usually employed to derive the approximation for the mean response time [15,23,24].

The approximation of tail latency for homogeneous FJQNs with phase-type service time distributions was introduced in a recent work [25], which is based on the analytical results from single-node and two-node networks. In [26], the authors proposed to use a black-box approach for the approximation of tail latency at high-load regions, treating each processing node as a black-box. The approach requires only the means and variances of response times at the processing nodes as input. However, it was based on the homogeneous assumption. Therefore, in this research, an analytical approach was derived for use in the introduced inhomogeneous FJQNs.

## 3. An Overview of Prediction Model

There is a recognized fact about the behavior of task flow, in parallel systems, that are waiting in lines for their turn to execute. Mainly, for any G/G/m queuing model that is highly loaded, it is observed that the accumulated tasks' waiting times behave as if they were independent variables following a distribution. The waiting times result from the build-up variability in task executions processed in front of the waiting lines and, obviously, in addition to the known differences in task arrivals. Further, it was noticed that

the distribution of the waiting times at a given line converged to a memory-less exponential distribution [27,28]. Following this convention, it posited that task sojourn times could be approximated to a distribution too. Hence, it is assumed that the resulting fluctuations from variability in service times are going to be alleviated, due to the impact of having long queues.

Consequently, on the basis of extreme value theory, request response times can be efficiently approximated if applying the central limit theorem in the sense that the two moments of historically collected response times for a single line, i.e., mean and variance, can be used to predict tail latency of response times using any suitable distribution. Having mentioned that, the generalized exponential distribution was determined such that it can capture different behaviors for the arrived jobs requests, e.g., heavy-tailed and short-tailed behaviors with regard to response times [12]. The presented results in [26] showed that the generalized exponential distribution outperformed exponential distribution in the matter of predicting tail latency. The following is the generalized exponential distribution function:

$$F_T(x) = (1 - e^{-x/\beta})^\alpha, \quad x > 0, \, \alpha > 0, \, \beta > 0, \tag{1}$$

distribution parameters $\alpha$ and $\beta$ are shape and scale, respectively. The mean and variance of the task response time are given [29]:

$$\mathbb{E}[T] = \beta[\psi(\alpha + 1) - \psi(1)], \tag{2}$$
$$\mathbb{V}[T] = \beta^2[\psi'(1) - \psi'(\alpha + 1)], \tag{3}$$

where $\psi(.)$ and its derivative are the digamma and poly gamma functions.

### 3.1. Prediction: Homogeneous

At a given fork-join structure, i.e., cluster of nodes, incoming users' requests are sent to a scheduler and forked to sub-requests/tasks that are mapped to the cluster's nodes. Now, assuming an approximated response times distribution at homogeneous conditions, the yield is the derivation that can be used for tail prediction:

$$F_X(x) = \prod_{i=1}^{k} F_{T_i}(x), \tag{4}$$

$$x_p = F_X^{(k)^{-1}}(p/100). \tag{5}$$

$$x_p = -\beta \log(1 - (\frac{p}{100})^{\frac{1}{k\alpha}}) \tag{6}$$

where $x_p$ is the $p$th percentile request response time; $\beta$ and $\alpha$ parameters can be obtained from the measured mean and variance at a single node, Equations (2) and (3). $k$ is the number of forked tasks, $1 \leq k \leq N$-nodes.

### 3.2. Prediction: Inhomogeneous

The experiments carried out on all previous works were based on the homogeneous assumption. Many cases were studied, and even cases for consolidated applications were covered. However, none of the works was enough to demonstrate the prediction performance when tested under non-homogeneous situations. Hence, putting the final piece of the puzzle into this ongoing research, a necessity to define ways to establish inhomogeneous conditions (i.e., HFJQNs) is direly needed for simulation experiments.

For the sake of clarity, consider a white-box approach and assume distinct distribution(s), either single or a mixture of distributions, associated with each node in a given fork-join system. Assume only one of many conditions causes heterogeneity in the fork-join structure to be applied during the whole simulation period, and let G reference the condition type that could exist. Heterogeneous conditions in this research, generally, are

driven by one of the following: (1) node's distribution $f(x)$, (2) node's utilization $\rho$, and/or (3) the tail of distribution applied at each node $\tau$.

Now, suppose there are $N$ mixtures, of $m$ distributions, at each of $N$ nodes: $SH_1$, $SH_2$, ..., $SH_N$. Let us assume that the distributions resulted from combined effects of both software and hardware. It is possible that the mixture at each node/computing unit might be uniquely different, i.e., $f_{j,i}(x) \neq f_{j,i+1}(x)$, for $i =$ from 1 to $N$-nodes, as defined in the following:

- Service times $x_j$, for $j = 1, 2, \ldots, m$, follow $f_j(x)$ distribution, where $x_j \sim f_j(x)$ with $\mu_j$ and $\sigma_j^2$.

- Assume each distribution $f_j(x)$ at node $i$ has weight $p_j$ in the mixture $SH_i$, and assume that for a long-run experiment, if using uniform distribution to maintain weights $p'_j s$, the resulted distributions are balanced. The effective random distributions are used to generate task service times:

$$X_i \sim f_i(x) = \sum_{j=1}^{m} p_{ij} \cdot f_{ij}(x). \tag{7}$$

The $k$th moment of $X$ at any fork node $i$ can be written as

$$\mathbb{E}[X_i^k] = \sum_{j=1}^{m} p_{ij} \cdot \mathbb{E}[X_{ij}^k]$$

$$= \sum_{j=1}^{m} p_{ij} \cdot \mu_{ij}^{(k)} = \mathbb{E}[\mathbb{E}[X_i^k | G_i]], \tag{8}$$

where $\mu_j^{(k)}$ is the $k$th moment of $X_j$, and G refers to one of the previously defined types of heterogeneous conditions that could exist at node $i$. Here, we assume only one for simplicity but in reality, it could be more complex.

According to the law of total variance, the variance of $X$ at fork node $i$ is given by

$$\sigma_i^2 = \mathbb{E}[Var(X_i | G_i)] + Var(\mathbb{E}[X_i | G_i])$$

$$= \sum_{j=1}^{m} p_{ij} \cdot \sigma_{ij}^2 + \sum_{j=1}^{m} p_{ij} \cdot \mu_{ij}^2 - \left( \sum_{j=1}^{m} p_{ij} \cdot \mu_{ij} \right)^2 \tag{9}$$

Since tasks spawned to the subsystems came from a large pool of many independent requests/applications, their *arrivals* at the system can be modeled using the Poisson process. Several studies have found that the dispersion around the inter-arrival-times distribution is very small, i.e., $C.V. = 1$ to 2, which means using the memory-less exponential is a reasonably acceptable model for simulating the arrivals of network traffic and data center applications [30–32]. Therefore, in this research, each fork node is viewed as an M/G/1 queue system, i.e., a Poisson arrival process with a general service time distribution and one service center.

The first and second moments of tasks waiting time at any fork node, i.e., an M/G/1 queuing system using Takács recurrence theorem [33], are given as follows:

$$\mathbb{E}[W_i] = \frac{\lambda \mathbb{E}[X_i^2 | G_i]}{2(1 - \rho_i)} = \frac{\rho_i \mathbb{E}[X_i | G_i]}{1 - \rho_i} \left( \frac{1 + C_{X|G}^2}{2} \right), \tag{10}$$

$$\mathbb{E}[W_i^2] = 2\mathbb{E}[W_i]^2 + \frac{\lambda \mathbb{E}[X_i^3 | G_i]}{3(1 - \rho_i)}, \tag{11}$$

where $C^2_{X|G} = \sigma^2_i / \mu^2_i$ is the squared coefficient of variation with $\sigma^2_i$ being the variance given by Equation (9); and $\rho_i = \lambda \cdot \mu_i$ is the utilization or load on $i$th Fork node at average arrival rate $\lambda$.

Therefore, from task executions' mean Equation (8) and variance Equation (9), respectively, the mean and variance of response times of the target task at $i$th Fork node can be written as

$$\mathbb{E}[T_i] = \mathbb{E}[W_i] + \mathbb{E}[X_i|G_i], \tag{12}$$

$$
\begin{aligned}
Var(T_i) &= Var(W_i) + Var(X|G), \\
&= (\mathbb{E}[W^2_i] - \mathbb{E}[W_i]^2) + (\mathbb{E}[X^2_i|G_i] - \mathbb{E}[X_i|G_i]^2),
\end{aligned}
\tag{13}
$$

where $\mathbb{E}[X|G]$ and $\mathbb{E}[X^2|G]$ are the first and second moments of target task service times, and $\mathbb{E}[W]$ and $\mathbb{E}[W^2]$ are given in Equations (10) and (11). The remainder is straightforward using Equation (5) and knowing the required moments for all of the $N$-nodes.

## 4. Definitions and Case Scenarios

Thinking logically, any possible inhomogeneous condition, per node in a fork-join structure, is mainly influenced by several elements. In this research, those elements are categorized into two primary factors: software and hardware. Many real scenarios can represent the software side, for example, operating systems, resources management, queuing management, utilization level, frameworks, daemons, ..., etc. While the hardware capabilities can be represented by differences in computing power, memory speed, hard disk, I/O, ..., etc., both sides participate in shaping any inhomogeneous situation at DCs. Hence, it is presumed that in a DC, the noticed differences in computing capabilities among serving units are tangible. This is due to the influence of both the resources' hardware and competing software, where their footprints on waiting times, queues, and task executions are clearly witnessed. Therefore, from the perspective of a primary cluster component, i.e., computing units, an abstract description is laid out to provide distinct realizations of how HFJQN could form. Each node behaves in a different way, as if it has a distinct distribution associated with it, or as the collective mixture of distributions applied at each node is unique.

We will demonstrate the simulation experiments considering per-node behavior (or a unique group of nodes) following unique distribution. Several defined perspectives are used in modeling the heterogeneous simulations, where examples from real-world scenarios are linked and used to explain some of the following abstract definitions. Therefore, experiments were conducted assuming a cluster of (1) different loads utilization among nodes, (2) the same utilization level for all nodes but different distributions' tails, (3) defined portions of homogeneous and inhomogeneous nodes [5] (93% of the Google cluster is homogeneous, which renders the remaining as inhomogeneous), and finally (4) with the help of multi-cores and multiprocessing technologies, different architectural designs per node applied, e.g., different numbers of replicas, distribution type, and mean service time. For the latter, as each node functions independently of others, we speculate that this case scenario might become the key kernel for any future simulations targeting heterogeneous designs. Basically, it is done by allowing each node to be handled independently as a separate event-based simulation process, where a recipient fork node treats each encapsulated task arrived from the system level as a newly arrived job at the subsystem level.

### 4.1. Different Load Utilization among Nodes

Load imbalance among nodes might result from many arising factors. For example, it happens due to combined operational effects of unbalanced scheduling policies (e.g., random dispatching [1]) and/or running multi-degree of applications fanouts. Additionally, it might occur due to the existing various computational powers of computing machines, the prolonged effects of prioritization and resource reservations, or as a result of emerging hotspots, where particular items/entities become popular, and the requesting

workloads for specific hotspots increase. Such unevenness in loads among a cluster's nodes could induce, in the long run, many distinguished levels of utilization. Therefore, to establish a similar scenario and compose HFJQNs rendering a clear difference in node utilizations, an exponential distribution is selected for such a case, but different mean service times are put to use, where the max one is selected to maintain the flow of the arriving requests into the system.

The several provided experiments with clusters consist of groups of 18 distinguished nodes, i.e., heterogeneous nodes, $h$-nodes = 18, where each one is uniquely utilized against others. The exponential distribution is the only applied distribution in this case, but with a different mean service time enforced at each node. Varied versions of utilization for the primary selected mean service time = 70 ms, which is the one utilized for the 90%, end up creating a range of several mean service times for all provided $N$ nodes at a given cluster. Experimented clusters are synthesized with the different load utilizations, starting at 5% and ending with 90% utilizations and applying 5% incremental steps in between.

Following the white-box approach, Table 1, we assume the same distribution is applied at all nodes, but different utilization ($\rho_i$) at each, where $i = 1$ to $N$ nodes. Hence, estimating the needed two moments for all the $N$-nodes of the fork-join structure can be easily done using Equations (13) and (12). The remainder is straightforward when using Equation (5) with the required moments for the $N$-nodes. The prediction results are presented in Figures 2 and 3. Refer to discussions regarding all the experimental results of this section as well others, which will be explained in more detail in Section 4.

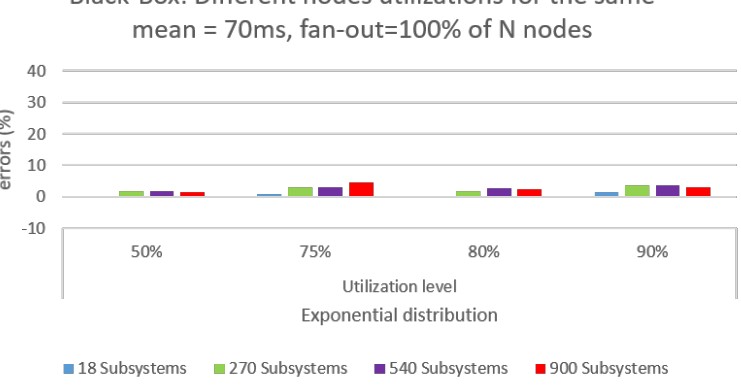

**Figure 2.** HFJQNs of different node utilizations, where the black-box approach is used in solution, and the load level (here) is determined by max node utilization.

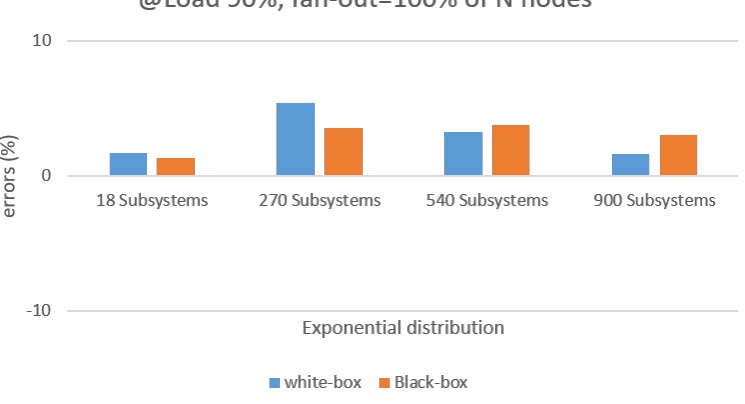

**Figure 3.** HFJQNs of different node utilizations, where the white and black-box methods prediction performances are compared, given the max node utilization is at load 90%.

**Table 1.** HFJQNs of different nodes utilizations.

| Parameters | $p_{ij} = 1, m = 1, G = \text{Utilization } (\rho)$ |
|:---:|:---:|
| Arrival Distribution | Poisson, $E[X\|\rho = 90\%]$ = 70 ms |
| Service Distribution(s) | Exponential, $E[X_i\|\rho_i]$ |
| Manipulated at $node_i$ | $\rho_i =$ in [5% *to* 90%], Step $\rho = 5\%$ |
| # Heterogeneous Nodes | $h$-nodes = 18 (or multiples of $h$-nodes) |
| Cluster Sizes | 18, 270, 540, 900 |

*4.2. Variant Distributions Tails among Nodes*

At high-load regions, where lots of incoming application requests are in waiting lines, contention for shared resources mounts, not to mention the diverse computing powers of resources that could exist among nodes. This might exhibit distinct differences in the overall tail response times at each node. Therefore, the combined effects of both, i.e., the increasing amount of applications demands, and the noted varieties in cluster hardware, can be presented in such a case scenario, where each node is bounded by a different tail $\tau_i$ than other nodes. Consequently, to demonstrate the effectiveness of the prediction model, examples of HFJQNs use multiple distributions with different tails, and each node is presumably associated with one of those several-tailed distributions.

In the following experiments, simulations with clusters consist of groups of nine distinct tails for nodes, i.e., heterogeneous nodes, $h$-nodes = 9, where each one has a unique distribution tail different from others, see Table 2. Heavy-tail Pareto distribution with the same mean service time, $\mu = 4.22$ ms, and lower bound L = 2.14 ms is assumed for all. The array of the nine tails, Table 3, originated by manipulating the shape parameter $\alpha_i$, which they used to distinguish each node in a single group of nine-tailed nodes. When the cluster scales up, it will become a multiple of the same single nine-node group.

Following the white-box approach, Table 4, we assume that the same distribution is applied at all nodes, the same utilization level as other nodes, but different tail ($\tau_i$) at each, where $i = 1$ to $N$ nodes. You can solve the complete FJ structure knowing the required moments for the $N$-nodes from using Equations (13), and (12) and then applying Equation (5). Prediction results are presented in Figures 4 and 5. Discussions regarding all the experiment results of this section as well as others will be explained in more detail in Section 4.

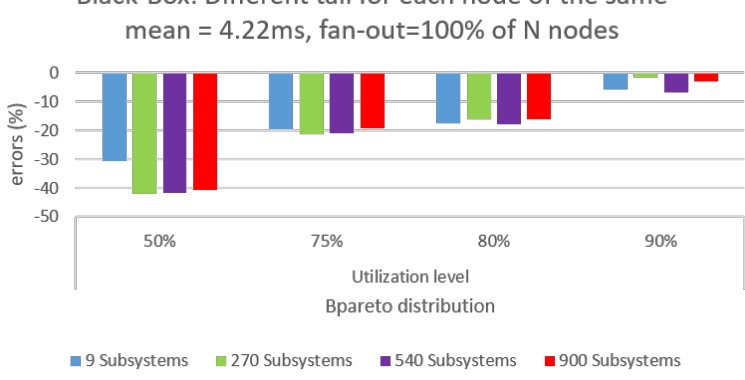

**Figure 4.** HFJQNs of nine distinct tails associated with all fork nodes, where the black-box method is used in the solution.

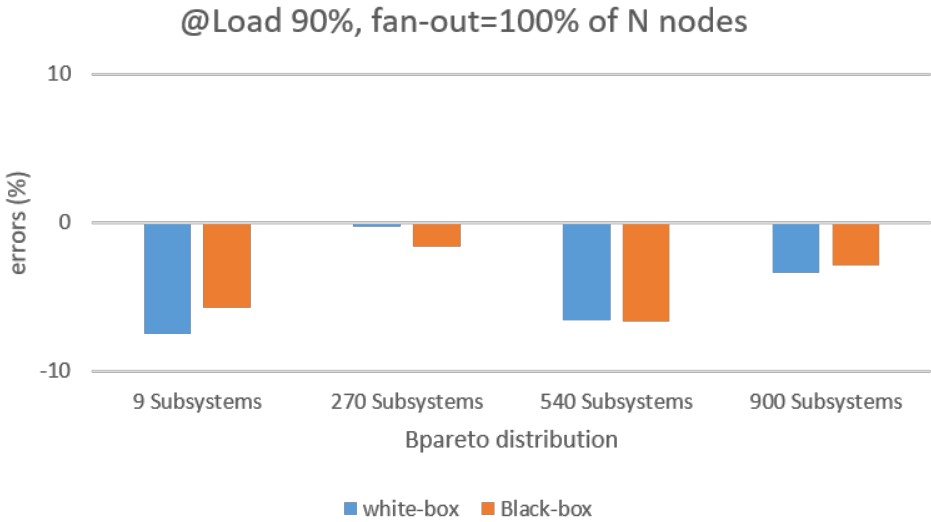

**Figure 5.** HFJQNs of nine distinct tails applied at all fork nodes, where the white- and black-box methods' prediction performance are compared at load 90%.

**Table 2.** HFJQNs of nine tails associated with all nodes.

| Parameters | $p_{ij} = 1, m = 1, \text{G} = \textbf{Tail } (\tau)$ |
|:---:|:---:|
| Arrival Distribution | Poisson, $E[X] = 4.22$ ms |
| Service Distribution(s) | Truncated Pareto, $E[X_i|\tau_i]$ |
| Manipulated at $node_i$ | $\tau_i$ = see Table 3 |
| # Heterogeneous Nodes | $h$-nodes = 9 (or multiples of $h$-nodes) |
| Cluster Sizes | 9, 270, 540, 900 |

**Table 3.** Pareto: different tails ($\tau_i$) for different shapes ($\alpha_i$).

| Distribution Tail $\tau_i$ | Shape Parameter $\alpha_i$ |
|:---:|:---:|
| 503.53 ms | 2.0210 |
| 450.00 ms | 2.0200 |
| 402.65 ms | 2.0189 |
| 350.07 ms | 2.0173 |
| 301.53 ms | 2.0153 |
| 250.39 ms | 2.0123 |
| 200.59 ms | 2.0078 |
| 150.51 ms | 2.0000 |
| 100.09 ms | 1.9833 |

*4.3. Separate Portions of Homogeneous and Inhomogeneous Nodes*

A straggler is a well-known phenomenon that might temporarily occur when service providers scale up their systems to cope with the growing need for extra resources. Even if a current state of a system is kept intact, there would be a potential chance for one or multiple serving units to slow down their executions of the incoming tasks. This might happens due to (1) spikes in requests for memory access or in CPU activity during tasks processing, (2) interference, in accessing storage servers or from irrelevant network traffic, (3) maintenance activities, such as garbage collections, data reconstructions, or (4) energy management and so on. All could induce similar situations for having a group of straggler

computing stations exists at any given cluster. For example, it was found in several studies on production traces of the Google cluster that a realized significant division of the cluster is homogeneous while the remaining division is inhomogeneous [5]. This fact is the basis of the subsequent experiments.

Assume a 90% of the given cluster nodes associated with exponential distribution, for $\mu = 4.22$ ms, and considered the homogeneous part. The remaining nodes, i.e., 10%, are left to be handled by the truncated Pareto distribution with max tail (upper bound) $= 503.530$ ms. We will follow through this example setup, and in another experiment, this ratio will be tuned as depicted in Figure 6.

For the white-box approach, Table 4, the same utilization level is enforced at all fork nodes. We assume two different distributions $f_i(x)$ associated with two groups of defined nodes' percentages, i.e., *h*-nodes = 90%(homogeneous) + 10%(in-homogeneous), where $i = 1$ to $N$ nodes. For solving the fork-join structure of $N$-nodes, refer to Equations (13) and (12) to estimate the required moments for all $N$-nodes, and then use Equation (5). The prediction results are presented in Figures 7 and 8. Discussions regarding all the experiment results of this section as well as others will be explained in more detail in Section 4.

In related scenario experiments, explained in Figure 6, we push this case a little further and observe the impact when tuning the portion of inhomogeneous nodes. Thus, the changing effects on jobs mean response times, the 99th percentile of the actual experiments response times, and the predicted 99th percentile tail-response times were studied, and the resulting outcomes are presented as shown. Truncated Pareto distribution of the same tail (i.e., 503.53 ms), and exponential distribution with mean execution times = 4.22 ms were applied. The outcomes of this experiment are no different than the basic one, where the errors in prediction stay the same, even knowing that the situation changed completely to homogeneous at the two ends of the figure, i.e., totally dominated by exponential distributions or Pareto distributions.

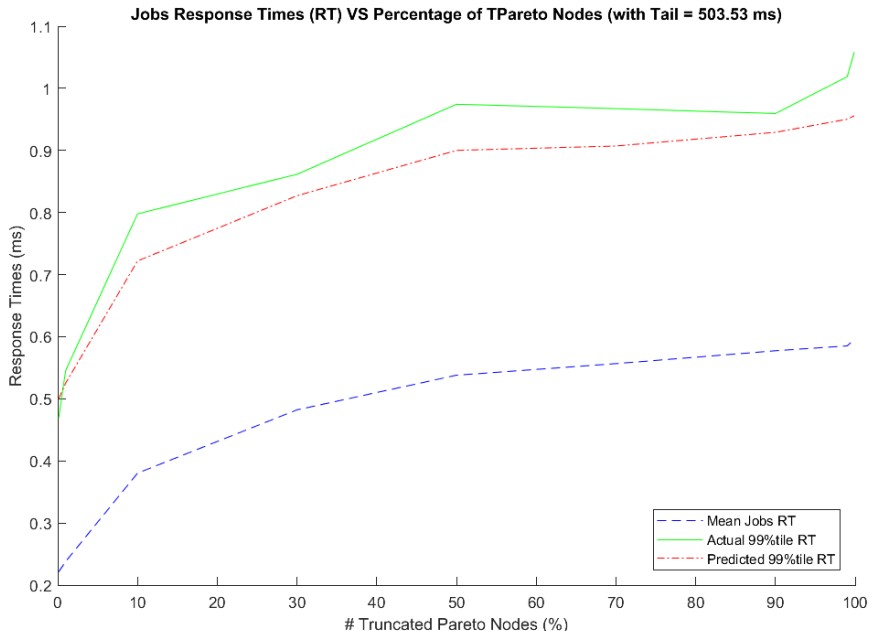

**Figure 6.** Job response times (RTs) against the increased percentage of inhomogeneous nodes (bounded Pareto with tail = 503.53 ms, and total cluster size = 900 nodes).

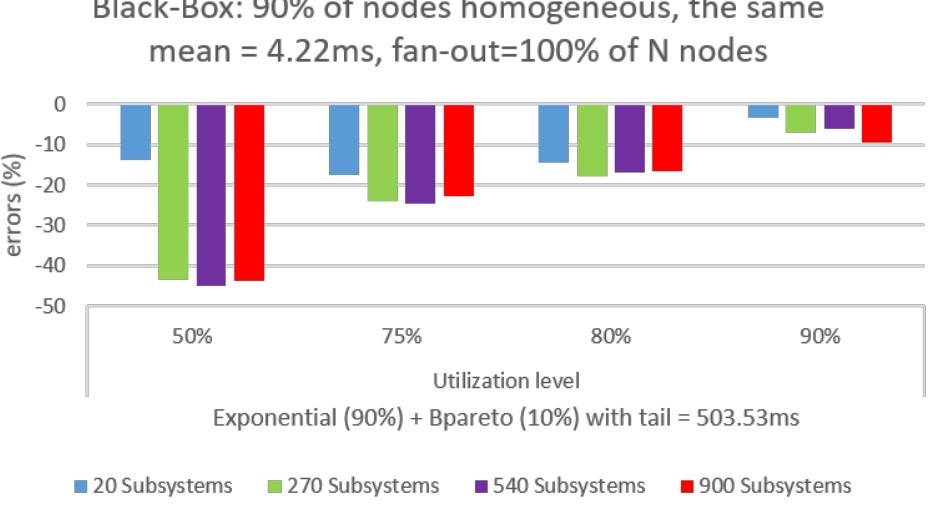

**Figure 7.** HFJQNs of applying 90% homogeneous (exponential) and 10% inhomogeneous (bounded Pareto) for all fork nodes, where the black-box method is used in the solution.

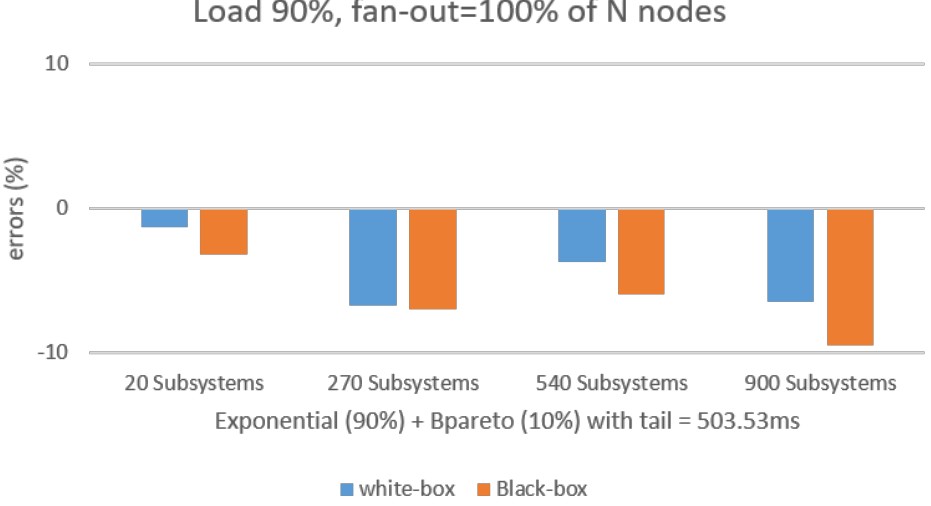

**Figure 8.** HFJQNs of applying 90% homogeneous (exponential) and 10% inhomogeneous (bounded Pareto) for all fork nodes, where the white- and black-box methods' prediction performance results are compared at load 90%.

**Table 4.** Portions of homogeneous and inhomogeneous nodes.

| Parameters | $p_{ij} = 1, m = 1, G = $ **Distribution** $f(x)$ |
|:---:|:---:|
| Arrival Distribution | Poisson, $E[X] = 4.22$ ms |
| Service Distribution (s) | TPareto and Exponential, $\mu = 4.22$ ms |
| Manipulated at $node_i$ | $f_i(x) = $ Expo or Pareto ($\tau = 503.53$ ms) |
| # Heterogeneous Nodes | $h$-nodes = 90% Expo + 10% Pareto |
| Cluster Sizes | 20, 270, 540, 900 |

*4.4. Nested Event-Based Simulations Model for Multi-Cores Technology*

A proposed scenario for heterogeneous modeling is to deviate from the usual practice in implementing event-based simulations and construct a simulation model of different architectural designs. For example, different node structures concerning replicas, distribu-

tions, utilizations, and even for establishing huge gaps between mean executions times, all can apply in a single round if using such simulation models. Multicore technology and a multiprocessing technique both were considered at the design stage to help exploit the suggested modeling scenario to the limits. Therefore, it became the sole motivation for conducting the following experiment, as it was desired to test the results of the prediction model for a situation where different node structures exist.

A vector of twenty-six distributions synthesized initially from nine distinct distributions, ten different mean service times, and nine Pareto distributions of different tails set with the same mean service time, $\mu = 4.22$ ms, were used for the experiment. The elements from the vector are assigned randomly to all nodes following uniform random distribution. A range of random replicas numbers is set arbitrarily for each node. Thus, diverse architectural designs are enforced per node. The Pareto distribution is used to generate target requests based on a predefined percentage to the system scheduler, where tagged forked tasks are sent to subsystems. Each subsystem is handled as different event-based simulation processes, where the tasks of the untagged job sent to null-event are used to advance the sub-simulation time. All sub-simulations function independently with varying configurations of their networks, and the number of generated task requests at each node is set to be unlimited. The created task requests are considered for the influence of having background applications involved. Recognizing the behaviors of background applications helped to study their impacts on target tasks received from the control system, i.e., from the parent simulation; see Figure 9.

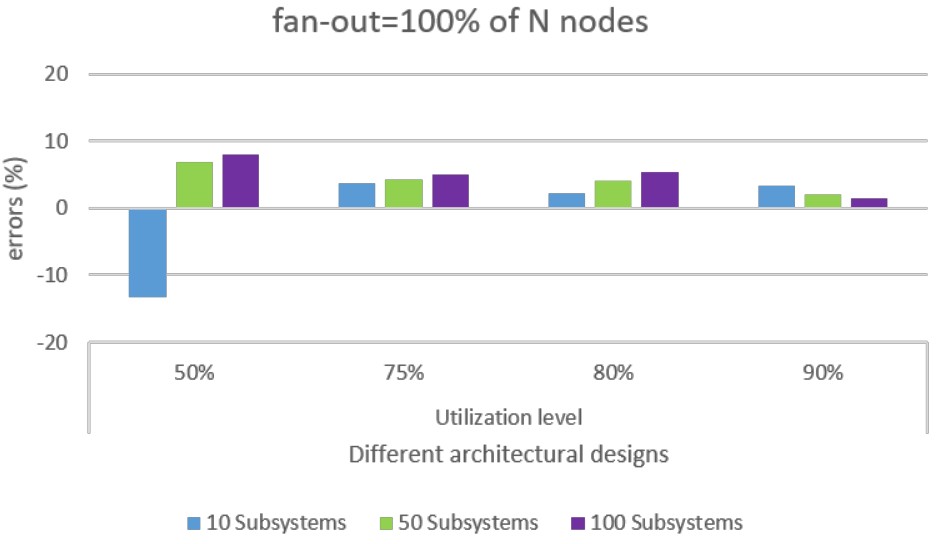

**Figure 9.** HFJQNs of applying different architectural designs at all fork nodes, where the black-box method is used in the solution.

## 5. Results and Discussion

To our knowledge, no one so far has exploited tail prediction in heterogeneous environments the way we did in this research work. Additionally, all of the existing state-of-the-art prediction models are known for targeting only homogeneous FJQNs, which leads to no available model for accuracy comparisons. Therefore, we used both approaches of our model—the white- and black-box methods—for comparing prediction results where it applicable.

For the white-box approach, to relax the complexities in computing the root, we even the number of nodes in all experiments groups, where the heterogeneous nodes of an experiment group, *h*-nodes, have different computing capabilities, e.g., unique distributions, or mean service times. Then it becomes a multiple of the same *h*-nodes as the system scale out to large cluster sizes, e.g., 270, 540, and 900. An exception is for the last case

scenario, where the *h*-nodes have different characteristics configurations per each scaled-out experiment, and that explains why only the black box is presented in that case.

Further, we numerically compute the root using the Levenberg–Marquardt algorithm, as the root-finding package comes by default with Matlab fsolve function. Applying the root-finding function with this algorithm requires assuming an initial value to begin with. From observations, it is found that this initial value has its impact and at the same time dictates the accuracy of tail prediction results. Hence, we conducted a small research study to observe the proper range of initial values for all heterogeneous conditions to determine one common range that would work under the selected root-finding package. From our analyses, we find that the right initial value for the utilization experiments using light-tailed distribution (e.g., exponential) is any value ($r_{init}$) that falls within the range $0.99 \leqslant r_{init} \leqslant 3.05$. For the remaining tests types, either applying the heavy-tailed Pareto of different tails or in combination with other distributions, the best initial value is $0.24 \leqslant r_{init} \leqslant 0.47$. The latter range also worked well with the heterogeneous cases covered by another research work conducted in parallel to this one, where mixtures of distributions are applied at each node. Actually, it is not clear whether this observation of range effect is due to the Matlab root-finding package or the estimation method, needing further research outside the scope of the current study.

For the types of HFJQNs in Section 3, except the last, Figures 2–8 present the prediction errors for the 99th percentile response times at loads 50%, 75%, 80%, and 90%, for all HFJQNs of N nodes that are running a single serving unit at each. The odd figures, i.e., Figures 3, 5 and 8, present comparisons of prediction errors between the black-box HFJQNs and the white-box HFJQNs at load 90% only. The model yields quite accurate approximations for tail latency at high load regions, less than 10% and 20% at the load of 90% and 80%, respectively. Further, in Figure 2, for all the cases with the exponential distribution, i.e., different node utilizations, the model gives accurate predictions across the multiple HFJQNs sizes studied. This is consistent with earlier observations i.e., the lighter the tail, the smaller the prediction errors. The results shown in the odd figures, i.e., the outcomes of comparisons between white box and black box, ideally, should be equivalent. The differences are introduced due to simulation and measurement errors, i.e., based on 95% confidence. For the example case with the nested simulations utilizing multi-cores technology in Section 4.3 the current implementation code relies heavily on memory, where the experiments are limited to small cluster sizes, i.e., 10, 50, and 100, and that is due to the limited resources of using a single core7-CPU testing PC. Figure 9 shows the prediction errors at the determined load levels, where, to our surprise, the prediction errors are reduced with less than 10% mostly across all load regions and less than 5% for loads at 75% or above. This further asserts our postulation about the reliability of using this model, assuming no advanced knowledge of node distributions, and even under such heterogeneous scenarios where huge differences in performance are applied among node architectures.

Lastly, one of the main questions that has not been answered yet is the considerable difference in prediction errors between heavy and light-tailed workload distributions. It seems the model successfully predicts the light-tailed distributions within the expected window of errors across all load levels. At the same time, it underestimates the heavy-tailed distributions at low loads, and once it reaches high loads, it overestimates, even though it stays within the expected margin of error.

As known, there are two competing effects in the approximation: (1) the impact of using extreme value theory, and (2) modeling our solution using a generalized exponential distribution. First, the solution is not an exact estimate but approximated; we use extreme value theory, which predicts the worst-case scenario. That explains why there is an expected small percentage of errors added to the approximation. Second, the basic concept is that at high-load regions, due to the queuing effect, the distribution of task response times of a single forked node can be modeled as an exponential, as its *CV* becomes near exponential in its behavior/characteristics, i.e., *CV* in the range [1–1.2]. Third, the used generalized expo-

nential model is inherently exponential, where the long tail can be considered as an added feature. Therefore, it will not have trouble predicting distributions of the short-tailed family at most load regions as we saw with the Weibull and the exponential distributions [34]. However, for heavy-tail distributions, and based on the concept, at high-load regions, when queuing effects dominate the existing workload distribution, 10% of errors in prediction, observed in all applied scenarios of workload distributions, is considered enough for the model to satisfy the argument.

## 6. Convergence Point for Reliable Metrics

The two performance metrics are essential for the work of the prediction model, and both are used to interpret any node's ability to serve a demanded latency response. Historical collections of task-completion times help in inferring the two metrics and determining the resource capabilities as well. Therefore, to gain better prediction, a crucial point is to learn the least amounts of archived responses, which is necessary for assessing the reliability of the metrics' measurements at a node. For example, assuming homogeneous conditions, it is known from the budget translation section that a single round of translation for a provided target response helps achieve the required nodes' measurements. Hence, $\mu$ and $\sigma^2$ can be obtained when numerically determining the value of C ratio, and inferring both $\alpha$ and $\beta$. Now, assuming a 10% window of errors in prediction is acceptable and a required 95% confidence level, Chebyshev's inequality can apply alongside the statistical metrics gathered from cluster nodes:

$$P(|X - \mu| > \varepsilon) \leqslant (1 - confidence) \tag{14}$$

where, $\varepsilon$ is the accepted error for the mean of task responses, i.e., $\varepsilon = (Accepted\ Error\ Ratio \cdot \mu)$. Now, assuming collected samples are independent, *central limit theorem* can be used for further derivation to normalize the $\sigma_{\bar{x}}^2$ of the population distribution into $\sigma^2/n$ for collected samples, as the following:

$$\frac{\sigma_{\bar{x}}^2}{\varepsilon^2} = \frac{\sigma^2}{n \cdot \varepsilon^2} \tag{15}$$

Therefore, we have

$$P(|X - \mu| > \varepsilon) \leqslant \frac{\sigma^2}{n \cdot \varepsilon^2} \tag{16}$$

From both Equations (14) and (16),

$$n = \frac{\sigma^2}{\varepsilon^2 \cdot (1 - confidence)} \tag{17}$$

Figure 10 demonstrates an example where the parameters were empirically gathered from carrying on two different types of experiments used for the estimation. The obtained statistics are from conducted homogeneous experiments [12], where mean and variance in responses were observed only for tagged tasks at workers. The used clusters were of different sizes (e.g., 10, 100, and 500 workers), and the collected responses of target tasks were based on predetermined target percentages (e.g., 10%, 50%, and 90%). Hence, the influence of having background applications was involved in the experiments. In the first scenario, the background was of the same distribution but applying a different mean service time than the target, and in the second one, the background was of a different distribution. The result in the figure shows that if a particular application gets to dominate a large percentage of the entire workload, the number of needed samples slightly increases $\leq 20\%$, where both two ends stay within the expected 10% of the target if set as a reference point when it dominates half of the workload. That is, once considered the addition of this slight amount, it does not matter how significant the portion of a target application will be. We follow the above derivation to mathematically estimate the required amount of samples for the same target tail once translated into $\mu$ and $\sigma^2$. The results are found to

be about the same with an accuracy of more than 87.5% compared with the one estimated from measurements. It is worth mentioning that the observed differences in the introduced errors caused by the average $\mu$ might be due to simulation and measurement errors, which could also negatively play a role in the changed portions of the target shown in the figure.

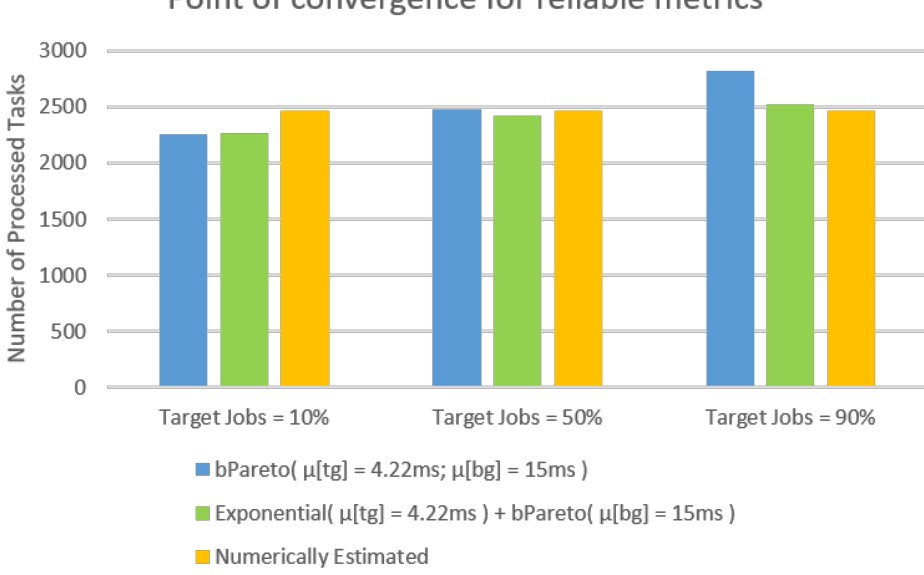

**Figure 10.** Samples amount required for reliable metrics, check scenarios 2 and 3 in [12].

Further, the amount of the required samples can be tuned for a lesser amount once the accepted error ratio of the mean latency is increased from the 10% shown in the figure to a little higher (e.g., 20%). Therefore, this method is a very crucial tool, as it presents a way to estimate the required amount of sample response times for getting reliable *mean latency* measurements of task processing at a fork node. Finally, using this approximation approach for estimating the required sample size is unique in a way that it does not need to know the distributions or nodes' statistical performance beforehand [35]. At the same time, it does not require to use approximations, which incur a high margin of errors when neither distributions nor nodes' measurements are known [36,37].

## 7. Budget Translation

One of the unique and potential benefits of using the prediction model is the ability to translate any required latency into task performance budgets, e.g., $\mu$ and $\sigma^2$. Consequently, from the given allowed budget, any landed job request negotiates for a range of nodes based on the knowledge of subsystems performance metrics that could serve while guaranteeing SLO requirement satisfaction. For that reason, a question of how to work out the budget translation using this model should arise. From Equations (2), (3) and (5), we already have the following:

$$\mu_T = g(\alpha, \beta) \tag{18}$$

$$\sigma_T = h(\alpha, \beta) \tag{19}$$

$$x_p = f(\alpha, \beta, p, k) \tag{20}$$

where $\mu_T$ and $\sigma_T$ are the mean and standard deviation of the task responses, and $x_p$ is the required tail in response times. Now, assume ratio $C = \mu_T / \sigma_T$, where $C \geqslant 1$. This

provides us with four equations and four variables We can numerically solve it, and hence, the translated budget will be

$$B_T = \mu_T \pm \sigma_T \tag{21}$$

For demonstration purposes, assume a homogeneous condition with the given user requirements: 99% ile, $x_p = 100$ ms, and amount of forked tasks $k = 20$. Applying the above method leads to the results shown in Table 5, where the translated budget can be used to direct tasks as in Figure 11. Choosing a $C$ ratio $\approx 1$ sounds reasonable to attain the performance measurements of a single node. A core principle of this model approximation is that at a high-load region, waiting times for a single node follow an exponential distribution, which means $\mu$ and $\sigma = 1/\lambda$. Therefore, it is safe to assume that for budget translation, setting $C$ ratio $= 1$ would provide us with approximately the same $\mu$ expected at any single node.

Accordingly, what happens if the $C$ ratio changes to different values? For example, assume changing the $C$ ratio takes the range between 1 to $\infty$. Figures 12 and 13 demonstrate the impact of tuning the $C$ ratio at the translated budget. Note that the observed changes in the boundaries stop when $C = 15$, but the increase in $\mu$ and the decrease in $\sigma$ values continue until $C = 31$. This means that working different $C$ ratios helps relax (low $C$) or tighten (high $C$) the amount of provisioned resources for the allowed budget.

**Table 5.** Translated budget using two different $C$ ratios.

| K | Latency | C | $\mu_{SLO}$ | $\sigma_{SLO}$ | Upper limit $_{SLO}$ | Lower limit $_{SLO}$ |
|---|---------|---|-------------|----------------|----------------------|----------------------|
| 20 | 100 ms | 1 | 13.16 ms | 13.16 ms | 26.32 ms | 0 |
|    | 100 ms | 15 | 60.46 ms | 7.22 ms | 67.68 ms | 53.23 ms |

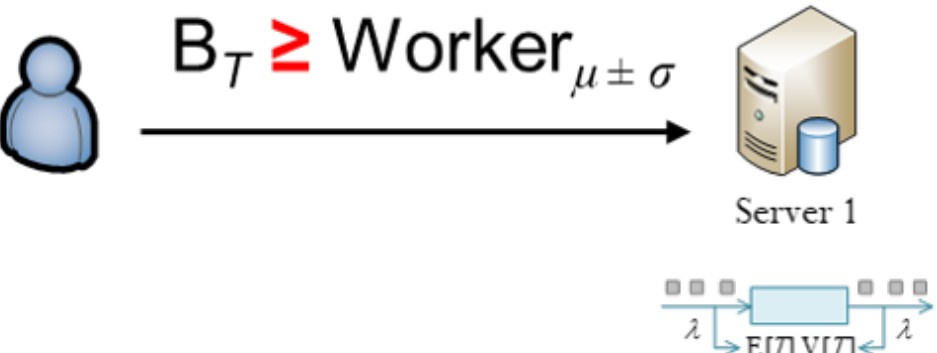

**Figure 11.** Using the translated budget to determine the right destination.

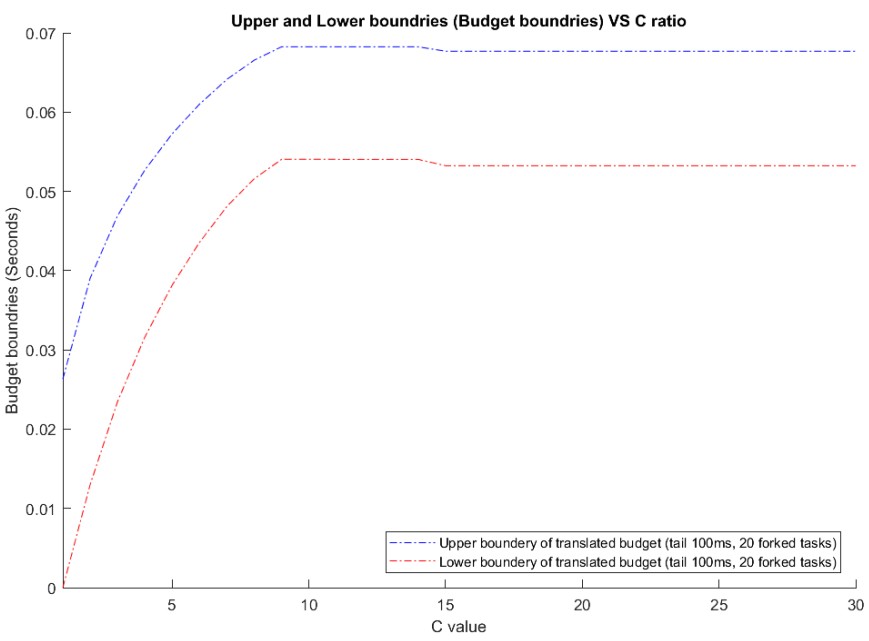

**Figure 12.** The two boundaries of the translated budget.

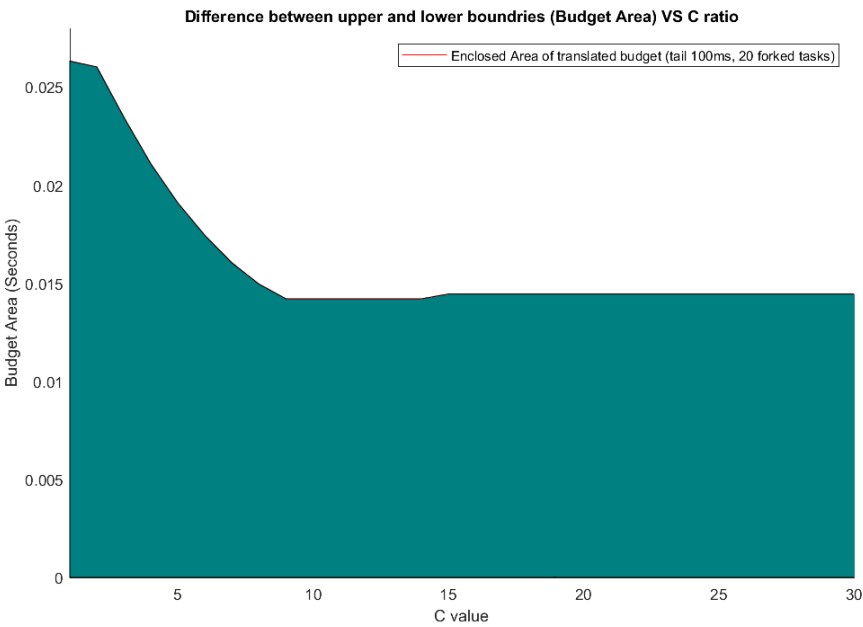

**Figure 13.** Translated Budget area: the enclosed area between the two boundaries.

## 8. Future Work and Conclusions

So far, provided there is a clear distinction between the inhomogeneous and homogeneous environments, the original-derived prediction model still holds. At schedulers, if decision making is established only based on queue lengths, it could sometimes be deceiving [1]. Henceforth, engaging the knowledge of the computing capabilities for cluster nodes to derive the resource allocation is more reasonable for gaining reliable performance. Schedulers would become able to define budget boundaries for guaranteed user satisfaction. Based on the performance measurements feedback, dispatching systems can specify ranges of nodes that could afford to serve users' requests within the given SLO requirements. A good example is to perceive an experienced taxi-driver who takes advantage of the

knowledge he has about a destination. He uses the knowledge to ride his vehicle and steer the wheel toward proper addresses. This is a crucial improvement that we could achieve from using the prediction model, which seems already to be lacking in other models.

In this paper, we deviated from the normal practice of most previously conducted works. Assuming heterogeneous conditions existed, we provided several study cases regarding the prediction of tail latency. Many conditioned scenarios were covered, where different distributions and unique characteristics were considered for the demonstration of cluster nodes. The model has proved to be reliable, as it performed well within an acceptable window of errors for all studied cases. The encouraging outcomes stand for adopting this prediction model in designed schedulers for improved decision making and providing guaranteed services. Furthermore, it is expected that this model could fit rightly in more sophisticated scenarios, where multi-stage scheduling is acquired [38]. Therefore, extending the use of the prediction model is the next future work, as the design nature of many production environments requires including more processing stages instead of only one fork-join phase.

**Author Contributions:** Conceptualization, S.A. and H.C.; methodology, S.A.; software, S.A.; validation, S.A., S.M. and H.C.; formal analysis, S.A.; investigation, S.M.; resources, S.M.; data curation, S.A.; writing—original draft preparation, S.A.; writing—review and editing, S.M.; visualization, S.A.; supervision, H.C.; project administration, S.M.; funding acquisition, S.M. All authors have read and agreed to the published version of the manuscript.

**Funding:** This research received no external funding.

**Institutional Review Board Statement:** Not applicable.

**Informed Consent Statement:** Not applicable.

**Data Availability Statement:** Not applicable.

**Conflicts of Interest:** The authors declare no conflict of interest.

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
