# Peer review of "Tail Prediction for Heterogeneous Data Center Clusters"

_processes, doi:10.3390/pr11020407_

Round 1

Reviewer 1 Report

The paper has discussed several techniques for modeling clusters of in-homogeneous nodes.

The Design includes nested-event-based simulation model borrowing help from multi-core technologies.

The works focuses on the tail latency can be predicted at high load regions.

I appreciate the contributions focused on budget translation in chapter 6. 

The authors are informed to clarify on the following.

1. Describe the process followed to select Case-Scenarios in chapter 3. (Clarify regarding this process: For the latter, as each node function independently from others, we speculate this case scenario might become the key kernel for any future simulations targeting heterogeneous designs.) 

2.  Regarding the results, kindly clarify how tail prediction in heterogeneous environments is unique? 

3. The authors are informed to justify how the model successfully predicts the light-tailed within the expected window of errors across all load-levels.

Author Response

Kindly find the word file attached

Regards

Reviewer 2 Report

1. For self-citation, several papers published by the second and third authors (Sami Alesawi and Hao Che) are cited in References 12, 13, 14, 15, and 22. Such a number of self-citation behaviors should not appear in academic papers, especially the above literature appearing in the literature review section is not appropriate. In addition to the relevant findings of the authors, are there other relevant studies that could prove that the generalized exponential distribution can capture different behaviors for the arrived jobs requests? It is necessary for the authors to review and sort out in detail. Meanwhile, the authors are suggested to replace some of the self-cited papers with works from other authors and supplement similar or related research results to support the findings of this paper better.

2. The paper structure needs to be remodeled. The significance of the authors’ design to show the results of association research in Chapter 7 is not clear. The related works should be summarized before the substantive research work is carried out. It is more appropriate to integrate the content of this chapter with the literature review.

Author Response

(The authors gave the same response as above.)
